# Variational Visual Question Answering for Uncertainty-Aware Selective Prediction

**Tobias Jan Wieczorek**                                      *tobias.wieczorek@tu-darmstadt.de*
*TU Darmstadt & hessian.AI, Germany*

**Nathalie Daun**
*TU Darmstadt & hessian.AI, Germany*

**Mohammad Emtiyaz Khan**
*RIKEN Center for Advanced Intelligence Project, Tokyo, Japan*

**Marcus Rohrbach**
*TU Darmstadt & hessian.AI, Germany*

**Reviewed on OpenReview:** *https: // openreview. net/ forum? id= jtnMIbJIso*

## Abstract

Despite remarkable progress in recent years, Vision Language Models (VLMs) remain prone to overconfidence and hallucinations on tasks such as Visual Question Answering (VQA) and Visual Reasoning. Bayesian methods can potentially improve reliability by helping models *predict selectively*, that is, models respond only when they are sufficiently confident. Unfortunately, such approaches can be costly and ineffective for large models, and there exists little evidence to show otherwise for multimodal applications. Here, we show for the first time the effectiveness and competitive edge of variational Bayes for selective prediction in VQA. We build on recent advances in variational methods for deep learning and propose an extension called "Variational VQA". This method improves calibration and yields significant gains for selective prediction on VQA and Visual Reasoning, particularly when the error tolerance is low ($\leq 1\%$). Often, just one posterior sample yields more reliable answers than those given by models trained with AdamW. In addition, we propose a new risk-averse selector that outperforms standard sample averaging by considering the variance of predictions. Overall, we present compelling evidence that variational learning is a viable option to make large VLMs safer and more trustworthy.

## 1 Introduction

Advances in VLMs (Wang et al., 2023; 2024; Li et al., 2024) have led to substantial gains on classical VQA benchmarks (Antol et al., 2015; Goyal et al., 2017), with performance now approaching or surpassing human levels. However, even strong VQA models are miscalibrated, prone to hallucinations, and often confidently guess answers instead of expressing uncertainty (*cf.* Fig. 1). In short, these models do not have a good notion of confidence about their own knowledge. This shortcoming hinders their deployment in safety-critical domains such as medical diagnosis or assistance for the visually impaired. When a model is confronted with adversarial (Sheng et al., 2021) or unanswerable (Bigham et al., 2010) inputs, which are common in the real world, these issues become even more pronounced.

One way to improve reliability is to allow a model to abstain when it is unsure of its response. The selective prediction framework formalizes such abstentions (Chow, 1957; El-Yaniv and Wiener, 2010), where the main

Question: Does the pedestrian light say walk?

Correct answer: "No"

Figure 1: Despite recent performance gains, VLMs trained with popular optimizers like AdamW do not know when they are wrong. Our VarVQA approach uses posterior variances to help the model decide when to abstain. The result shown on the right is for BEiT-3 (Wang et al., 2023), which achieves near-human accuracy on VQAv2 (Goyal et al., 2017).

challenge is to find a good confidence estimator that separates correct answers from incorrect ones. Given such an estimator, the incorrect answers can be replaced with an output that indicates that the model does not know the answer such 'Not sure' or 'I do not know', which replaces potentially costly errors with abstentions. Although recent work has connected selective prediction to hallucinations, (Kalai et al., 2025), the literature on multimodal models remains relatively sparse. Previous approaches in the multimodal domain have proposed to incorporate additional model components to improve confidence estimates: Whitehead et al. (2022) train a lightweight head on top of the frozen VLM backbone, while Srinivasan et al. (2024) use external vision tools and an additional language model to quantify uncertainty. Both works do not attempt to improve the reliability of the underlying model. Instead, their solutions introduce additional overhead to the prediction pipeline while also adding new failure points for uncertainty estimation.

Variational Bayesian (VB) methods (Graves, 2011) can potentially address the unreliability of VLMs without requiring additional components or tools. In particular, the uncertainty in the learned posterior distribution over model parameters can be used to help the model make a prediction only when it is sufficiently confident. This theory remains untested though, as for a long time, VB approaches have been ineffective for large transformer-based architectures. However, the recently developed Improved Variational Online Newton (IVON) optimizer (Shen et al., 2024) has enabled effective variational training of models such as GPT-2 (Radford et al., 2019) with no significant overhead compared to AdamW (Loshchilov and Hutter, 2019). So far, IVON has been limited to unimodal domains, and in this work, we are the first to test its effectiveness for multimodal applications, particularly with regards to selective prediction. Our contributions are as follows:

1. We demonstrate that variational training is effective for large multimodal transformer-based architectures and introduce the Variational VQA (VarVQA) framework for selective VQA abstentions.

2. We demonstrate improved uncertainty estimation across multiple dimensions: better calibration, and enhanced selective prediction with particularly large gains at low error tolerances, as well as increased robustness under distribution shift.

3. We establish superior sample efficiency compared to Monte-Carlo (MC) Dropout, showing that Variational VQA provides better reliability given an equal compute budget.

4. We propose a new risk-averse selector function that leverages output variance, yielding consistent improvements in *high-stakes* selective prediction where errors are particularly costly.

## 2 Background and Related Work

### 2.1 Uncertainty and Reliability in Visual Question Answering

Visual Question Answering (VQA) is a popular multimodal task that requires a model to understand two modalities and their interaction to predict answers. While multimodal models (Li et al., 2023; Wang et al., 2023; 2024) have recently achieved human-level performance on standard benchmarks like VQAv2 (Goyal et al., 2017), they still perform poorly at selective prediction on the same benchmarks (Dancette et al., 2023). Our work here is the first to address the selective prediction task using variational Bayesian methods.

**Selective Prediction.** In the selective prediction framework (Chow, 1957; El-Yaniv and Wiener, 2010), a "selector" assigns a confidence score to the answer given by a model and subsequently decides whether the prediction is accepted or the model is forced to abstain instead (that is, it says "I do not know"). This decision is made by comparing the assigned confidence against a given abstention threshold. In VQA in particular, the model learns a function $f : \mathcal{I} \times \mathcal{Q} \to \mathcal{A}$ to predict an answer $a \in \mathcal{A}$, given a multimodal input $x = (i, a)$ consisting of an image $i \in \mathcal{I}$ and a question $q \in \mathcal{Q}$. In selective prediction notation, the model output space is augmented by an *abstain* output $\emptyset$. This transforms the predictive model $f$ into a selective model $m$, incorporating both $f$ and a selector $g$. The answer $f(x)$ is accepted if $g(x)$ is above the abstention threshold $\gamma \in \mathbb{R}$, and rejected otherwise. We follow the notation of Whitehead et al. (2022):

$$m(x) = (f, g)(x) = \begin{cases} f(x) & \text{if } g(x) \geq \gamma, \\ \emptyset & \text{if } g(x) < \gamma. \end{cases} \tag{1}$$

A high threshold $\gamma$ corresponds to a conservative case, in which the model answers only the questions on which it is most confident. Lowering $\gamma$ reduces abstentions (higher coverage), but increases the error rate among accepted answers (higher risk). The tradeoff between risk and coverage is unavoidable, but a better confidence estimator yields a lower risk at any given coverage level. In practice, the cost of error or the risk level is specified in advance, and $\gamma$ is set accordingly, see Section 4.2. Typically, the answer likelihood (Geifman and El-Yaniv, 2017) or the predictive entropy are used as selection functions.

Most of the prior work on selective prediction can be classified into either external approaches or integrated approaches. In external setups, a selector is built on top of the frozen predictive model, for example in the form of a trainable model head (Whitehead et al., 2022; Mielke et al., 2022; Mushtaq et al., 2025), LoRA parameters (Chen et al., 2023) or vision tools (Srinivasan et al., 2024). In integrated setups, predictor and selector have at least one combined training phase. Integrated selectors take different forms as well, such as a model head (Geifman and El-Yaniv, 2019) or a dedicated abstention class (Ziyin et al., 2019). However, if model and selector are trained together, instabilities often ensue, which require special treatment (Geifman and El-Yaniv, 2019). Bayesian approaches have not been considered for selective prediction so far, with the exception of concurrent work by (Daheim et al., 2025), which has explored IVON for generative language modeling, although they do not consider multimodal tasks. In contrast to prior work on selective prediction in VQA, our objective is to *directly* improve the reliability of model confidence estimates without additional parameters, training phases, or tools. In other words, we train VLMs where reliability is "baked-in" by design, not added as an afterthought.

**Calibration.** Calibration represents a different angle on uncertainty estimation, namely the alignment of a model's predictive confidence with its accuracy: When the model expresses $x\%$ confidence in an answer, it should be correct $x\%$ of the time. The difference between calibration and selective prediction becomes clear when considering a model that is correct on $y\%$ of examples and is also always exactly $y\%$ confident (for a fixed $y$ satisfying $0 \leq y \leq 100$). Although this model is perfectly calibrated, it cannot distinguish its correct and incorrect outputs and thus fails at the task of deciding when to abstain. Prior work has found that large neural networks often exhibit overconfidence, particularly in Out-Of-Domain (OOD) settings (Snoek et al., 2019). In unimodal classification tasks, temperature (Guo et al., 2017) and Platt (vector) Platt et al. (1999)

scaling are effective at improving calibration. Ensembling (Lakshminarayanan et al., 2017) typically yields even better results, but requires prohibitive resources to train $N$ models. New ideas, such as prompting the model to express a verbalized confidence have been mostly ineffective for VLMs (Xuan et al., 2025). We show that Variational VQA yields well-calibrated VLMs, achieving a lower Expected Calibration Error (ECE) than vector scaling, while matching other sampling methods like Monte-Carlo Dropout (Gal and Ghahramani, 2016). In general, we argue that for a VLM to be reliable, it should be calibrated and also know when to abstain. Both these are improved with Variational VQA in comparison to models trained with the AdamW optimizer.

## 2.2 Variational Bayes for Deep Learning

Variational Bayesian Learning provides a principled approach to estimate uncertainty by learning probability distributions (often Gaussians) over the weights of a neural network. While conventional deep learning methods estimate network weights $\theta$ by minimizing *empirical risk* $\bar{\ell}(\theta) = \frac{1}{N}\sum_{i=1}^{N}\ell_i(\theta)$, *variational* methods estimate a distribution $q(\theta)$ over network weights by minimizing the variational objective

$$\mathcal{L}(q(\theta)) = \lambda\mathbb{E}_{q(\theta)}\left[\bar{\ell}(\theta)\right] + \mathbb{D}_{\mathrm{KL}}(q(\theta) \parallel p(\theta)). \tag{2}$$

Here, $N$ is the size of the training set, $\ell_i(\theta)$ the loss for example $i$, $\mathbb{D}_{\mathrm{KL}}(q\|p)$ denotes the Kullback-Leibler divergence, $\lambda$ is a scaling parameter, and $p(\theta)$ the prior weights distribution. To keep computational costs manageable, $q(\theta)$ is often chosen to be a diagonal-covariance Gaussian, that is, $q(\theta) = \mathcal{N}(\theta \mid m, \mathrm{diag}(v))$, where $m$ and $v$ are the parameter mean and variance vectors. The objective $\mathcal{L}(q(\theta))$ can be reparametrized in terms of $m$ and $v$, and $\mathcal{L}(m,v)$ is typically approximated through MC sampling of the weights.

**IVON.** In the early 2010s, variational methods that directly optimize parameter means and variances through standard deep learning techniques such as SGD achieved promising results (Graves, 2011; Blundell et al., 2015). However, in subsequent years, these approaches could not keep up with the growth in scale of network architectures (Trippe and Turner, 2018; Foong et al., 2020; Coker et al., 2022). Recently, natural gradient methods that build Hessian estimates through an Adam-like update (Khan et al., 2018; Osawa et al., 2019) have addressed this issue. The IVON optimizer (Shen et al., 2024) further develops those and obtains comparable accuracy and better uncertainty estimates than AdamW at nearly identical training cost. In particular, IVON uses an Adam-like (Kingma, 2014) update for the parameter means $m$ and variances $v$. Similarly to adaptive scaling in Adam, IVON updates $m$ by using gradients scaled with $h$ - an online estimate of the diagonal Hessian. In an update step, we first sample $\theta \sim q$, then compute a minibatch gradient $\hat{g}$. The following four lines are used to compute the minibatch Hessian estimate $\hat{h}$, update the moving average of the Hessian $h$, and then update the means $m$ and variances $v$:

$$\hat{h} \leftarrow \frac{\hat{g}(\theta - m)}{v}, \tag{3}$$

$$h \leftarrow \beta_2 h + (1 - \beta_2)\hat{h} + \frac{(1 - \beta_2)^2(h - \hat{h})^2}{2(h + \delta)} \tag{4}$$

$$m \leftarrow m - \alpha\frac{\bar{g} + \delta m}{h + \delta}, \tag{5}$$

$$v \leftarrow \frac{1}{\lambda(h + \delta)}. \tag{6}$$

Here, $\alpha$ is the learning rate, $\delta$ the weight decay and $\bar{g}$ a moving average of the gradient. IVON also uses Adam-like momentum for the gradients and the Hessian. A notable difference to Adam, however, is the absence of the square root over the scaling vector $h + \delta$ in line Eq. (5). Notably, the initialization of the

Hessian estimate $h_0$ is a crucial hyperparameter to set carefully. For more details, we refer to the original paper by Shen et al. (2024).

We use IVON because it offers several advantages compared to other Bayesian baselines. Unlike the Laplace approximation (MacKay, 1992; Daxberger et al., 2021), it does not require an additional pass through the data to compute the Hessian. Neither does it require additional training like Stochastic Weight Averaging (SWA) (Izmailov et al., 2018). Compared to MC Dropout (Gal and Ghahramani, 2016), the advantage is the availability of a fixed posterior form that can be more easily used for downstream tasks. For instance, the method is easily amenable to ensembling (Lakshminarayanan et al., 2017), which can further improve performance (Daheim et al., 2025). We offer new insights compared to previous IVON works (Shen et al., 2024; Cong et al., 2025; Daheim et al., 2025), by showing its effectiveness in training multimodal models and for selective prediction. We further propose a new selection function that uses the output variance, which was never utilized in prior work.

## 3 Variational VQA

Our Variational VQA approach uses the IVON optimizer to train large VLMs and evaluates the reliability of its output confidences in comparison to baselines like AdamW and MC Droput. In Section 3.1, we describe how model confidences are obtained, in Section 3.2 we describe the baseline selectors, and in Section 3.3 we present our new risk-averse selector.

### 3.1 Inference and Model Confidence

At inference, variational methods typically use the learned posterior through MC sampling. However, if computing efficiency is imperative, one can ignore the variances (set $v = 0$) and use the mean parameters $m$ for inference (Shen et al., 2024), which requires only one forward pass. We refer to this approach as 'VarVQA mean'. For an input $x$, denote the model's output likelihood vector by $f(x; \theta)$, where the k-th entry $f_k(x; \theta)$ contains the likelihood of class $k$ (out of $K \in \mathbb{N}$ classes). VarVQA, on the other hand, performs sampling, that is, we draw model parameters $\theta^{(s)} \sim q$ to get $S \in \mathbb{N}$ likelihood vectors $f(x; \theta^{(s)})$. These are aggregated to obtain a mean likelihood vector $\bar{\mu}$ and a mean likelihood variance vector $\bar{\sigma}^2$:

$$\bar{\mu}(x) = \frac{1}{S} \sum_{s=1}^{S} f(x; \theta^{(s)}), \qquad \bar{\sigma}^2(x) = \frac{1}{S-1} \sum_{s=1}^{S} \left[ f(x; \theta^{(s)}) - \bar{\mu}(x) \right]^2. \tag{7}$$

### 3.2 Baseline selector functions

We start with the baseline selector for deterministic methods (AdamW, VarVQA mean). We employ the widely used MaxProb selector (Geifman and El-Yaniv, 2017), which uses the answer likelihood. In a classification task, the MaxProb selector is defined as $g_{\mathrm{MP}}(x) = \max_k f_k(x; \theta)$. We use MaxProb, because we find it to consistently outperform predictive entropy and related functions. In the case of multiple samples, a reasonable baseline is predictive averaging (Gal and Ghahramani, 2016). Here the selector is $g_{\mathrm{MP}}^{\mu}(x) = \max_k \bar{\mu}_k(x)$, where $\bar{\mu}_k(x)$ is the $k$-th entry of the vector $\bar{\mu}(x)$, *cf.* Eq. (7).

### 3.3 A new risk-averse selector

In this work, particularly for the context of selective prediction, we propose to go Beyond Predictive Averaging (BPA) by also employing the empirical output variances (*cf.* Eq. (7)). This is done in a risk-averse (Pratt, 1978) manner, by penalizing high-variance predictions. While Pratt (1978) subtracts the variance (with a prefactor), we found the standard deviation to work best:

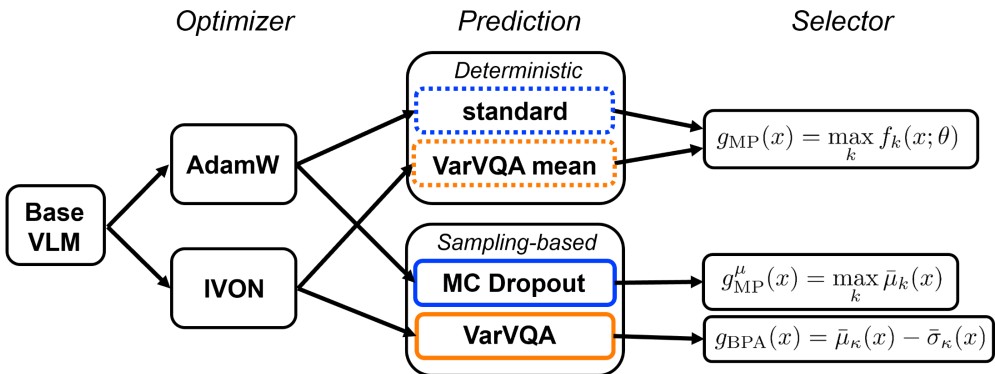

Figure 2: Overview of the methods we experiment with and their selectors. Variational VQA employs $g_{\mathrm{MP}}^{\mu}$ for calibration and $g_{\mathrm{BPA}}$ for selective prediction.

$$g_{\mathrm{BPA}}(x) = \bar{\mu}_{\kappa}(x) - \bar{\sigma}_{\kappa}(x) \qquad (8)$$

Here, $\kappa = \arg\max_k \bar{\mu}_k(x)$. Thus, the risk-averse selector does not change the prediction, only the confidence.

Another perspective motivating $g_{\mathrm{BPA}}$ comes from Bayesian credible intervals. If we approximate the distribution of the highest-likelihood class as Gaussian — that is, $\mathcal{N}(\bar{\mu}_{\kappa}, \bar{\sigma}_{\kappa}^2)$ — then the term $\bar{\mu}_{\kappa} - \bar{\sigma}_{\kappa}$ corresponds to approximately the 16th percentile of this distribution. In other words, there is roughly an 84% posterior probability that the true class probability exceeds $\bar{\mu}_{\kappa} - \bar{\sigma}_{\kappa}$. This provides a conservative lower confidence bound that is particularly valuable in high-stakes selective prediction where overconfident predictions carry significant costs. We find that the $1\sigma$ choice empirically balances conservatism against practical utility.

All our selective prediction results with VarVQA use $g_{\mathrm{BPA}}$. In Section 4.5, we provide an ablation against predictive averaging. When it comes to calibration, VarVQA uses predictive averaging, as the subtraction of $\sigma$ leads to systematic underconfidence[1]. When using MC Dropout with AdamW, we found no systematic benefits of $g_{\mathrm{BPA}}$. We speculate that this is because the posterior was not actively learned. Thus, we use only $g_{\mathrm{MP}}^{\mu}$ for Dropout. The selectors used for each method are visually summarized in Figure 2.

## 4 Experiments

We describe our experimental setup, models and datasets in Section 4.1 and the evaluation metrics in Section 4.2. Our results show that Variational VQA is effective for multimodal models, more sample-efficient than MC Dropout (Sec. 4.3), and more robust to OOD data than AdamW-trained models (Sec. 4.4). Moreover, our novel selector $g_{\mathrm{BPA}}$ outperforms posterior predictive averaging on high-stakes selective prediction (*cf.* Sec. 4.2) across multiple models and tasks (Sec. 4.5).

### 4.1 Experimental Setup

We explore the effectiveness of Variational VQA on two large VLMs: ViLT (Kim et al., 2021) and BEiT-3 (Wang et al., 2023). BEiT-3 is near-SOTA[2] on VQAv2, but still small enough for full fine-tuning. Both ViLT and BEiT-3 treat VQA as a classification task to 3129 answers, which is standard practice (Anderson et al., 2018). In terms of multimodal tasks, we explore VQA (fine-tuning on VQAv2 (Goyal et al., 2017), evaluation on VQAv2 and AdVQA (Sheng et al., 2021)) and Visual Reasoning (fine-tuning and evaluation on NLVR2 (Suhr et al., 2019)). The publicly available VQAv2 test splits do not include labels, which are

---

[1]In selective prediction, only relative confidences matter, so there is no negative impact.
[2]As of 10/2025, see the VQAv2 Challenge on EvalAI

required to evaluate calibration and selective prediction (*cf.* Sec. 4.2). Therefore, we follow previous work (Whitehead et al., 2022) and divide the VQAv2 validation set into dev/val/test. All results are averaged over three training runs with different seeds. Error bars and shaded regions indicate standard error.

**Hyperparameters.**   We use the optimal hyperparameters reported in (Kim et al., 2021; Wang et al., 2023) for AdamW. For IVON, most defaults, *cf.* Shen et al. (2024), can be used, but the learning rate and Hessian initialization need to be adjusted. However, we find that due to a strong correlation between the two, the dimensionality of the search space is effectively one. A full account is provided in Appendix A.

**Number of Samples.**   By default, Variational VQA uses $S = 64$ MC samples. We did not find significant improvements beyond this number. For early stopping, we use eight MC samples to save compute.

**Temperature and Vector Scaling.**   Previous work (Whitehead et al., 2022) has shown that calibrating models with widespread methods like Temperature Scaling (Guo et al., 2017) and Vector Scaling (Platt et al., 1999) has only a small effect on their selective prediction performance. We confirm these findings and show that the effect is consistently positive, and can be applied on top of any method (*e.g.* AdamW or VarVQA) to receive small additional gains. Full results are in Appendix C.

## 4.2   Evaluation Metrics

**Accuracy.**   We work with the standard VQA accuracy (Antol et al., 2015), which can also take non-integer values (0.3, 0.6, 0.9), besides 0 and 1, if fewer than 4 out of 10 annotators agree. NLVR2 accuracy is binary.

**Calibration.**   We evaluate calibration using the Expected Calibration Error (ECE) (Pakdaman Naeini et al., 2015; Guo et al., 2017), as is standard practice. The ECE is computed by partitioning the model's answer confidences on a dataset $\mathcal{D}$ into $m$ bins $\mathcal{D}_m$, and then summing the bin-wise deviations of confidence from accuracy. We use $m = 15$ in our experiments.

$$\text{ECE} = \sum_{m=1}^{M} \frac{|D_m|}{|D|} \left| \text{Acc}(D_m) - g(D_m) \right|. \tag{9}$$

**Coverage at Risk.**   For the selective prediction metrics, we follow prior work (Geifman and El-Yaniv, 2017; Whitehead et al., 2022; Dancette et al., 2023). The standard selective prediction metric is *Coverage at Risk* ($C@R$), which measures the percentage of questions the model is able to answer (*i.e.* where it does not abstain), while keeping the error tolerance $r$ below a given risk level $R$:

$$C@R = \max_{\gamma} C(\gamma) \quad \text{s.t.} \quad r(\gamma) \leq R, \quad \text{where we define} \tag{10}$$

$$C(\gamma) = \frac{1}{|D|} \sum_{x \in D} \mathbb{1}(g(x) \geq \gamma), \quad \text{and} \tag{11}$$

$$r(\gamma) = \frac{\frac{1}{|D|} \sum_{x \in D} (1 - \text{Acc}(f(x)) \mathbb{1}(g(x) \geq \gamma)}{C(\gamma)}. \tag{12}$$

$$\tag{13}$$

A larger $C@R$ is better, as a model that abstains on (almost) all inputs is not useful. We also compute the area under the Risk-Coverage curve (AUC) (Kamath et al., 2020). A weakness of $C@R$ is that the

threshold $\gamma$ is determined using the test set. This is necessary as otherwise, a comparison of results would be challenging: For a given risk $R$, one would have to judge both *threshold generalization* (*i.e.* whether the test risk matches the bound $R$), and the achieved test coverage.

**Effective Reliability.** Whitehead et al. (2022) suggested *Effective Reliability* $\Phi_c$ that avoids test set threshold selection. It differs from accuracy by a negative cost $c$ assigned to wrong answers:

$$\phi_c(x) = \begin{cases} \text{Acc}(x) & \text{if } g(x) \geq \gamma \text{ and } \text{Acc}(x) > 0, \\ -c & \text{if } g(x) \geq \gamma \text{ and } \text{Acc}(x) = 0, \\ 0 & \text{if } g(x) < \gamma. \end{cases} \tag{14}$$

The total effective reliability is $\Phi_c = \frac{1}{|\mathcal{D}|} \sum_{x \in \mathcal{D}} \phi_c(x)$, and the abstention threshold $\gamma$ is determined by optimizing $\Phi_c$ on validation data. We report accuracy (Acc), $C@R$ and $\Phi_c$ in percent, while keeping the ECE in $[0, 1]$ to be consistent with Whitehead et al. (2022).

**High-Stakes metrics.** Both selective prediction metrics ($C@R$ and $\Phi_c$) feature a parameter that controls the severity of mistakes. Our findings match previous work (*cf.* Tabs. 1,2 in (Whitehead et al., 2022)): Models disproportionately struggle with settings in which errors are very costly (low-$R$, high-$c$)[3]. We collectively refer to these metrics as *high-stakes*. For practical applications, it is arguably more important that models perform well in high-stakes metrics than in low-stakes metrics, since large amounts of errors (even as low as 5%) are not acceptable in many real-world scenarios. Moreover, for ID experiments we observe saturation[4] in low-stakes metrics and thus focus our reported results on high-stakes.

It should be noted that, if stakes are set too high (*i.e.* cost $c$ too high or risk $R$ too small), results can become noisy, as the impact of individual overconfident samples rises. This issue increases with smaller and less well-curated datasets (label noise can have an impact). In our experiments, we observe that the results were stable only up to $c \approx 100$ and down to $R \approx \frac{1}{2}\%$, which is why we stop reporting there.

### 4.3 In-Distribution Experiments

We show ID results after fine-tuning on VQAv2 in Table 1 and on NLVR2 (Visual Reasoning) in Table 2. Figure 3 visualizes the VQAv2 results. Variational VQA matches the accuracy of the conventional AdamW optimizer (Fig. 3a), while achieving better calibration in terms of lower ECE (Fig. 3b), and better (high-stakes) selective prediction in terms of higher $C@1\%$ (Fig. 3c) and $\Phi_{100}$ (Fig. 3d). Additionally, 'VarVQA mean' (*cf.* Sec. 3.1), is frequently more reliable than AdamW (lower ECE, higher $C@R$, $\Phi_c$), while needing the same inference compute. Finally, the VarVQA sampling strategy consistently outperforms MC Dropout, which uses the same amount of samples at inference, in terms of selective prediction, while achieving a low ECE of $\lesssim 0.03$ throughout and $< 0.02$ on VQAv2 with all three tested models. Regarding selective prediction, the benefits of VarVQA over AdamW are largest for the high-stakes metrics. When only one mistake per 200 samples is allowed ($C@\frac{1}{2}\%$), VarVQA on different VLMs improves $7\% - 9\%$ on VQAv2 and $9\% - 14\%$ on NLVR2 vs. AdamW. When Deep Ensembles (Lakshminarayanan et al., 2017) are applied on top of an existing method, the reliability improves (*cf.* Table 3). However, on VQAv2, even vanilla VarVQA is often better than the AdamW Ensemble and the VarVQA Ensemble stays consistently ahead. More in-depth results are in Appendix Appendix F. We note that Deep Ensembles cause significant overhead through the necessity of training $N$ models instead of one. (We use $N = 3$ due to computational constraints.)

---

[3]The achieved $C@R$ and $\Phi_c$ in these settings are much further below the theoretical optimum than for high $R$/low $c$.

[4]For example, BEiT-3 large on VQAv2 achieves $C@10\% > 81\%$ and $C@20\% > 98\%$.

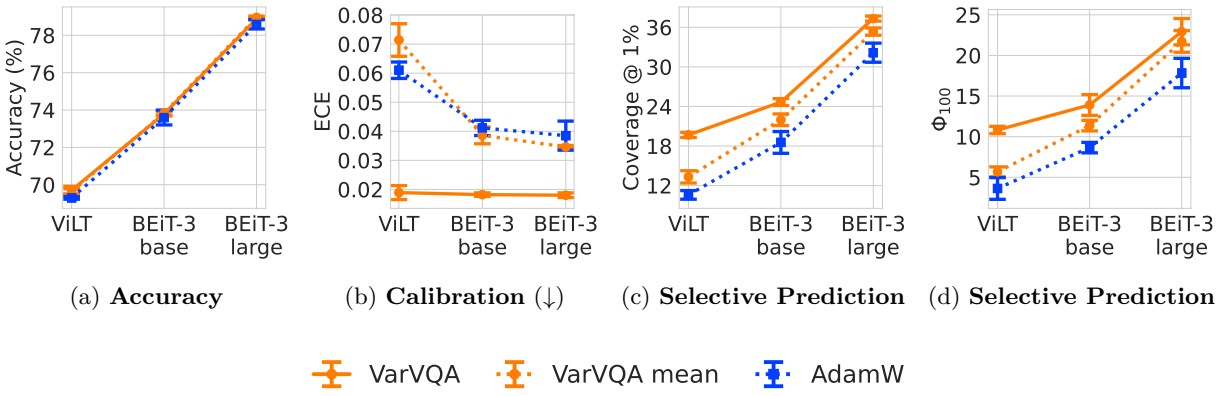

Figure 3: Accuracy, calibration and selective prediction results for different models after fine-tuning on VQAv2. Error bars indicate standard error across three seeds.

Table 1: Reliability evaluation on VQAv2 for fine-tuned models. The variable $N$ denotes the number of forward passes. Best results per model are **bold**.

| Model | Method | $N$ | Acc. | Calibration ECE ($\downarrow$) | Selective Prediction *high-stakes* $C@\frac{1}{2}\%$ | $C@1\%$ | $\Phi_{50}$ | $\Phi_{100}$ | Sel. Prediction *low-stakes* $C@5\%$ | $\Phi_{10}$ |
|---|---|---|---|---|---|---|---|---|---|---|
| ViLT | AdamW | 1 | 69.30 | 0.061 | 5.03 | 10.58 | 8.41 | 2.89 | 36.24 | 24.05 |
| | VarVQA mean | 1 | 69.63 | 0.071 | 6.77 | 13.32 | 9.74 | 5.45 | 37.93 | 25.08 |
| | AdamW Dropout | 64 | 69.66 | **0.019** | 10.44 | 16.63 | 12.51 | 8.44 | 38.49 | 26.18 |
| | VarVQA | 64 | 69.71 | **0.019** | **13.81** | **19.68** | **12.93** | **10.88** | **39.53** | **27.15** |
| BEiT-3 base | AdamW | 1 | 73.60 | 0.041 | 10.35 | 18.55 | 15.59 | 8.65 | 47.93 | 33.40 |
| | VarVQA mean | 1 | 73.84 | 0.039 | 14.08 | 21.98 | 16.72 | 11.36 | 49.57 | 34.80 |
| | AdamW Dropout | 64 | 73.46 | 0.019 | 13.07 | 20.11 | 16.61 | 9.44 | 47.49 | 33.36 |
| | VarVQA | 64 | 73.79 | **0.018** | **18.10** | **24.66** | **19.26** | **13.90** | **49.76** | **35.22** |
| BEiT-3 large | AdamW | 1 | 78.59 | 0.039 | 21.63 | 32.15 | 26.31 | 17.80 | 63.19 | 45.83 |
| | VarVQA mean | 1 | 78.96 | 0.035 | 25.32 | 35.35 | 28.31 | 21.25 | **64.83** | 47.43 |
| | AdamW Dropout | 64 | 78.41 | **0.018** | 25.28 | 34.52 | 27.99 | 20.65 | 63.00 | 46.23 |
| | VarVQA | 64 | 78.89 | **0.018** | **28.13** | **37.05** | **29.56** | **23.21** | 64.68 | **48.06** |

## 4.4 Mixed ID/OOD Experiments

Following (Dancette et al., 2023), we use VQAv2 (Goyal et al., 2017) and AdVQA (Sheng et al., 2021) as ID and OOD datasets, respectively. Both datasets use COCO images (Lin et al., 2014), but AdVQA has a different multimodal distribution (more challenging questions). We use the splits from (Dancette et al., 2023), which draw testing data from $P_{\text{mix}}$, where

$$P_{\text{mix}} = \alpha P_{\text{OOD}} + (1 - \alpha)P_{\text{ID}}, \tag{15}$$

using $P_{\text{ID}} = $ VQAv2 and $P_{\text{OOD}} = $ AdVQA. Different mixtures are obtained by varying $\alpha \in [0, 1]$. Figure 5 shows the results for BEiT-3 large. Although the accuracy drops equally fast for all methods, Variational VQA remains better calibrated (Fig. 5b). The decline in $C@1\%$ is equal in absolute numbers (Fig. 5c), but this implies that the relative performance of VarVQA vs. AdamW is increasing at higher OOD fractions.

Table 2: Reliability evaluation on NLVR2 for fine-tuned models. The variable $N$ denotes the number of forward passes. Best results per model are **bold**.

| Model | Method | $N$ | Acc. | Calibration | Selective Prediction high-stakes | | | | Sel. Prediction low-stakes | |
|---|---|---|---|---|---|---|---|---|---|---|
| | | | | ECE ($\downarrow$) | $C@\frac{1}{2}\%$ | $C@1\%$ | $\Phi_{50}$ | $\Phi_{100}$ | $C@5\%$ | $\Phi_{10}$ |
| BEiT-3 base | AdamW | 1 | 83.45 | 0.059 | 6.42 | 11.61 | 4.58 | 2.24 | 54.79 | 26.18 |
| | VarVQA mean | 1 | 83.28 | 0.058 | 5.15 | 15.58 | 6.44 | 1.41 | 55.66 | 27.30 |
| | AdamW Dropout | 64 | 83.18 | **0.016** | 9.98 | 15.99 | 6.95 | 2.95 | 55.43 | 27.63 |
| | VarVQA | 64 | 83.11 | 0.031 | **15.42** | **23.36** | **11.20** | **5.00** | **57.16** | **29.23** |
| BEiT-3 large | AdamW | 1 | 88.34 | 0.041 | 16.53 | 41.14 | 18.08 | 9.45 | 78.53 | 45.64 |
| | VarVQA mean | 1 | 88.83 | 0.062 | 17.15 | 31.07 | 15.27 | 3.57 | 80.17 | 45.02 |
| | AdamW Dropout | 64 | 88.11 | **0.017** | **33.21** | 44.69 | 23.43 | 14.71 | 76.99 | 46.55 |
| | VarVQA | 64 | 89.26 | 0.029 | 32.89 | **49.24** | **25.56** | **14.85** | **82.11** | **49.51** |

Table 3: Comparison of VarVQA to Deep Ensembles (Lakshminarayanan et al., 2017), applied to AdamW-trained models and on top of VarVQA; both on VQAv2. The ensembles use three models each.

| Model | Method | Acc. | Calibration | Selective Prediction high-stakes | | | | Sel. Prediction low-stakes | |
|---|---|---|---|---|---|---|---|---|---|
| | | | ECE ($\downarrow$) | $C@\frac{1}{2}\%$ | $C@1\%$ | $\Phi_{50}$ | $\Phi_{100}$ | $C@5\%$ | $\Phi_{10}$ |
| ViLT | AdamW Ensemble | 69.69 | 0.049 | 6.97 | 12.30 | 9.97 | 3.67 | 37.86 | 25.14 |
| | VarVQA | 69.71 | 0.019 | 13.81 | 19.68 | 12.93 | 10.88 | 39.53 | 27.15 |
| | VarVQA Ensemble | 70.08 | 0.018 | 14.39 | 20.09 | 14.88 | 10.65 | 40.51 | 27.57 |
| BEiT-3 base | AdamW Ensemble | 74.70 | 0.018 | 15.84 | 23.66 | 19.19 | 11.82 | 51.33 | 36.65 |
| | VarVQA | 73.79 | 0.018 | 18.10 | 24.66 | 19.26 | 13.90 | 49.76 | 35.22 |
| | VarVQA Ensemble | 74.18 | 0.015 | 18.34 | 25.70 | 19.27 | 10.98 | 51.16 | 36.01 |
| BEiT-3 large | AdamW Ensemble | 79.45 | 0.020 | 26.50 | 36.95 | 30.51 | 19.78 | 66.25 | 48.84 |
| | VarVQA | 78.89 | 0.018 | 28.13 | 37.05 | 29.56 | 23.21 | 64.68 | 48.06 |
| | VarVQA Ensemble | 79.14 | 0.015 | 28.68 | 37.97 | 30.40 | 23.91 | 65.56 | 48.52 |

Thus, there is reason to believe that Variational VQA may be fundamentally more robust to OOD data than AdamW-trained models. The results for the other models and metrics are in Appendix D.

### 4.5 Beyond Predictive Averaging

We compare the performance of our novel selector $g_{\text{BPA}}$ (*cf.* Sec. 3.3) to the baseline $g_{\text{MP}}^{\mu}$ (*cf.* Sec. 3.2). The full results are shown in Tables 4 and 5. For the high-stakes selective prediction metrics, $g_{\text{BPA}}$ consistently outperforms the sample averaging of $g_{\text{MP}}^{\mu}$, achieving *e.g.* 5% higher $C@\frac{1}{2}\%$ on NLVR2 for BEiT-3 base. For the mostly saturated low-stakes selective prediction metrics (grayed), there is no clear winner. When using MC Dropout, we did not find any systematic improvement of $g_{\text{BPA}}$ over $g_{\text{MP}}^{\mu}$.

### 4.6 Qualitative Results

We show qualitative examples that highlight the difference in uncertainty estimates between AdamW and Variational VQA in Figures 6 and 7. Further qualitative examples for VQAv2, AdVQA and NLVR2, including failure cases, can be found in Appendix E. As the accuracy of the AdamW- and IVON-trained models is

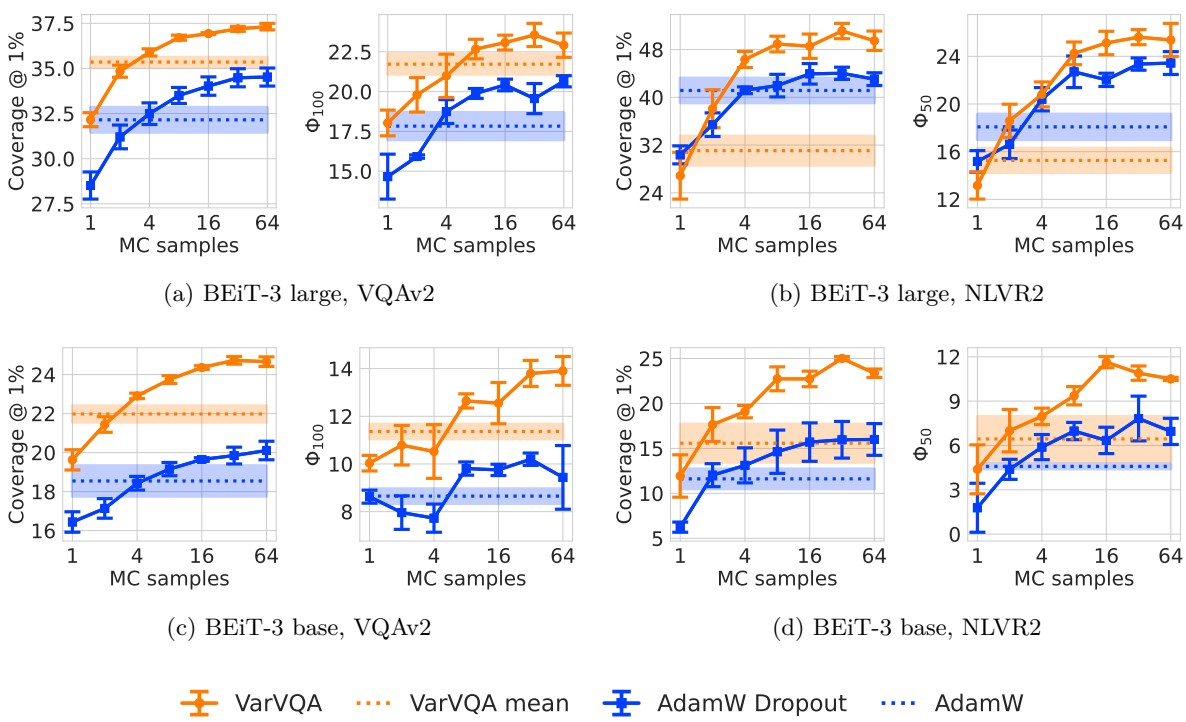

Figure 4: Comparison of Variational VQA to MC Dropout, which uses the same inference compute, on high-stakes selective prediction. Error bars and shaded regions indicate standard error across three seeds.

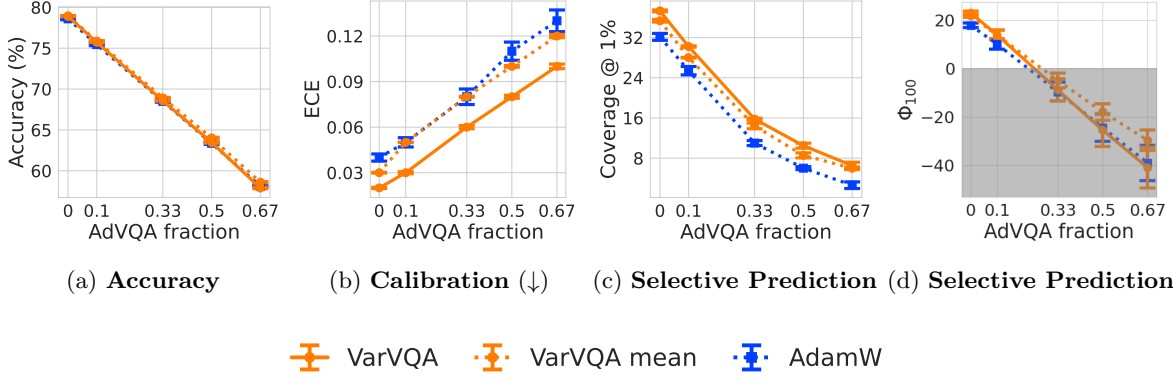

Figure 5: Accuracy, calibration and selective prediction results for different VQAv2/AdVQA mixtures for BEiT-3 large. Error bars indicate standard error across three seeds. In **(d)**, every model in the gray area is worse than a model that abstains on every input.

similar, we focus on cases where they predict the same answer, as this reflects the typical behavior. The key improvement of VarVQA lies not in better accuracy, but rather in improved uncertainty estimates. A further study that investigates the behavior on the different question categories of VQAv2 and AdVQA (*Binary*, *Number*, and *Other*), can also be found in Appendix E.

## 5 Discussion

In this work, we explore Variational VQA, *i.e.* the application of Variational Learning for multimodal tasks. Our implementation replaces the standard AdamW optimizer with the IVON method and uses multiple sam-

Table 4: Comparison of our risk-averse selection function $g_{\text{BPA}}$ (Eq. (8)) against $g_{\text{MP}}^{\mu}$ on VQAv2 with VarVQA ($N = 64$ samples as always). Best results per model are **bold**.

| Dataset | Model | Selector | high-stakes | | | | low-stakes | |
|---|---|---|---|---|---|---|---|---|
| | | | $C@\frac{1}{2}\%$ | $C@1\%$ | $\Phi_{50}$ | $\Phi_{100}$ | $C@5\%$ | $\Phi_{10}$ |
| VQAv2 | ViLT | $g_{\text{MP}}^{\mu}$ | 13.35 | 19.24 | **13.04** | 10.05 | 39.52 | 26.64 |
| | | $g_{\text{BPA}}$ | **13.81** | **19.68** | 12.93 | **10.88** | **39.53** | **27.15** |
| | BEiT-3 base | $g_{\text{MP}}^{\mu}$ | 17.15 | 23.87 | 18.64 | 12.23 | **49.91** | 35.17 |
| | | $g_{\text{BPA}}$ | **18.10** | **24.66** | **19.26** | **13.90** | 49.76 | **35.22** |
| | BEiT-3 large | $g_{\text{MP}}^{\mu}$ | 27.09 | 36.00 | 28.82 | 22.14 | **64.82** | 47.58 |
| | | $g_{\text{BPA}}$ | **28.13** | **37.05** | **29.56** | **23.21** | 64.68 | **48.06** |

Table 5: Comparison of our risk-averse selection function $g_{\text{BPA}}$ (Eq. (8)) against $g_{\text{MP}}^{\mu}$ on NLVR2 with VarVQA ($N = 64$ samples as always). Best results per model are **bold**.

| Dataset | Model | Selector | high-stakes | | | | low-stakes | |
|---|---|---|---|---|---|---|---|---|
| | | | $C@\frac{1}{2}\%$ | $C@1\%$ | $\Phi_{50}$ | $\Phi_{100}$ | $C@5\%$ | $\Phi_{10}$ |
| NLVR2 | BEiT-3 base | $g_{\text{MP}}^{\mu}$ | 10.64 | 22.20 | 9.75 | 3.95 | **57.18** | **29.28** |
| | | $g_{\text{BPA}}$ | **15.42** | **23.36** | **11.20** | **5.00** | 57.16 | 29.23 |
| | BEiT-3 large | $g_{\text{MP}}^{\mu}$ | 27.61 | 48.16 | 24.26 | 13.59 | **82.16** | **49.51** |
| | | $g_{\text{BPA}}$ | **32.89** | **49.24** | **25.56** | **14.85** | 82.11 | **49.51** |

ples from the learned posterior at inference. In addition, our new selector goes beyond the standard predictive averaging by incorporating the output's variance into the abstention decision. Our findings demonstrate that Variational VQA has two possible applications: when inference costs should be minimal, parameter means can be used at inference to at least match the accuracy of AdamW and decently increase reliability. When higher inference costs are acceptable, multiple MC samples from the posterior can be used. Better reliability is demonstrated by better calibration as well as better selective prediction, both in distribution for multiple tasks, and in the challenging mixed ID/OOD setting. The novel selector further improves selective prediction in high-stakes settings with almost no computational overhead.

Variational VQA also has some limitations, particularly involving hyperparameter tuning with IVON. While we observe correlations between the critical hyperparameters (discussed in the Appendix), which can be exploited to reduce the search space, tuning still remains more involved than with AdamW. Additionally, while VarVQA makes large gains in high-stakes selective prediction vs. AdamW, overconfidence still remains an issue, and Coverages remain well below the theoretical optimum ($\approx$ *Acc.* for low risks). Thus, more work is needed to make models truly 'know what they do not know'.

An exciting avenue for future work is to avoid the computational burden of sampling for VarVQA by variance propagation in one forward pass. Recently, Li et al. (2025) proposed a new method in this domain that has shown promising results for unimodal tasks with IVON. Such 'streamlining' is only possible if learned parameter variances are available, which is not the case for *e.g.* MC Dropout. While Variational VQA intrinsically improves reliability, the incorporation of previous methods through *e.g.* training a (variational) selector on top of the (variational) model, could also further enhance reliability. Improvements could also be obtained by using a more expressive posterior that uses full covariance, however at the time of writing, there is no practical alternative to IVON for large models that uses full covariance.

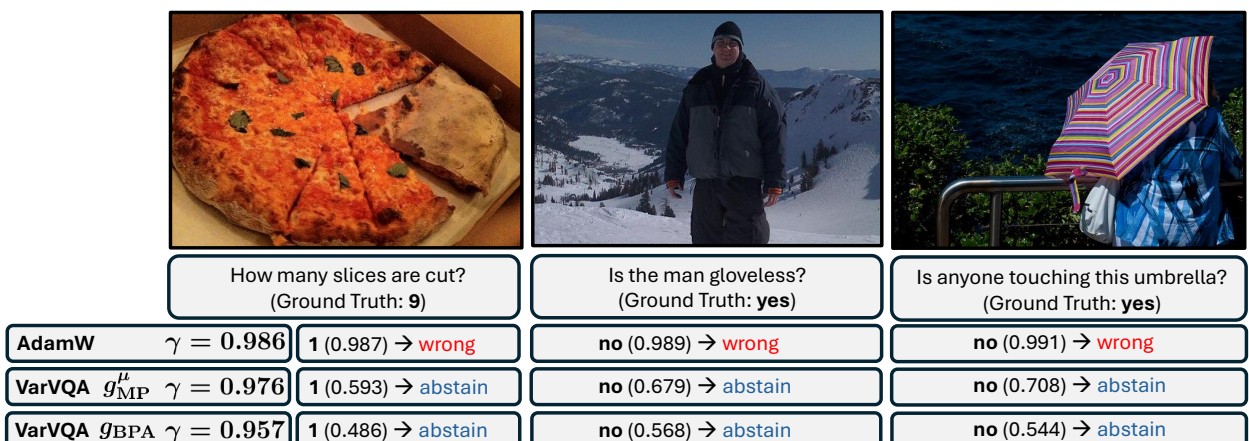

Figure 6: Qualitative examples on VQAv2 with BEiT-3 large where AdamW is wrong while VarVQA abstains. The abstention thresholds $\gamma$ were determined by optimizing $\Phi_{100}$ on VQAv2 validation data. Model answers are displayed in **bold**, the corresponding answer confidences are provided in brackets.

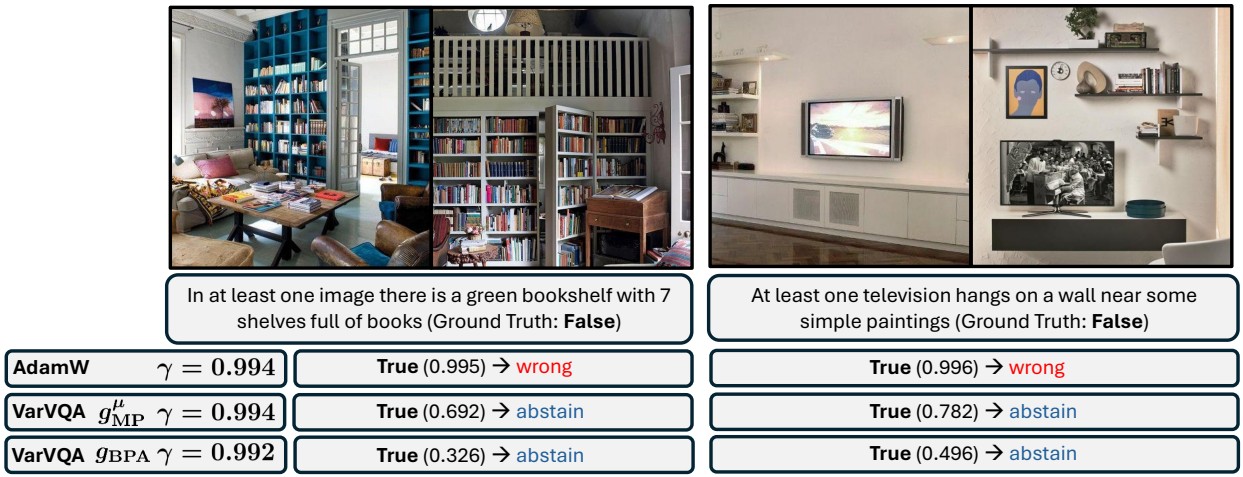

Figure 7: Qualitative examples on NLVR2 with BEiT-3 large where AdamW is wrong while VarVQA abstains. The abstention thresholds $\gamma$ were determined by optimizing $\Phi_{100}$ on NLVR2 validation data. Model answers are displayed in **bold**, the corresponding answer confidences are provided in brackets.

**Acknowledgements.** This research was partially funded by an Alexander von Humboldt Professorship in Multimodal Reliable AI, sponsored by Germany's Federal Ministry for Research, Technology and Space and by a LOEWE-Spitzen-Professur (LOEWE/4a//519/05.00.002(0010)/93). Mohammad Emtiyaz Khan was supported by the Bayes duality project, JST CREST Grant Number JPMJCR2112. The work has benefited from the Excellence Cluster "Reasonable AI" by the Deutsche Forschungsgemeinschaft (DFG, German Research Foundation) under Germany's Excellence Strategy – EXC-3057. For compute, we gratefully acknowledge support from the hessian.AI Service Center (funded by the Federal Ministry of Research, Technology and Space, BMFTR, grant no. 16IS22091) and the hessian.AI Innovation Lab (funded by the Hessian Ministry for Digital Strategy and Innovation, grant no. S-DIW04/0013/003).

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
