# OpenReview forum: "Variational Visual Question Answering for Uncertainty-Aware Selective Prediction"
_TMLR — Accepted by TMLR_

### Review · Reviewer_ZqNN · 2025-12-05

**Summary Of Contributions:**

This paper proposes Variational VQA (VarVQA), a framework for uncertainty estimation in multimodal vision–language models by training with the IVON variational optimizer, so that the model learns parameter variances that can be leveraged for uncertainty-aware selective prediction. The authors claim to be the first to apply IVON to VLMs, and they provide empirical evidence of improved uncertainty behavior.

A second contribution is a simple risk-averse selection rule that adjusts confidence using predictive dispersion—specifically, subtracting the (top-class) standard deviation from the mean confidence—which improves high-stakes selective prediction in their experiments.

Strengths. (1) The paper targets a clear and timely problem (overconfidence/hallucination and selective prediction in VQA). (2) Experiments across multiple backbones show comparable accuracy while improving high-stakes selective prediction, and often improving calibration. (3) The compute-controlled comparison against MC Dropout is a useful and practically relevant baseline.

Concerns / weaknesses. (1) The novelty may feel primarily empirical/system-level: IVON itself is not new, and the proposed selector is simple and lacks a strong theoretical connection to the selective-prediction objective beyond intuition. (2) IVON appears to introduce additional hyperparameters and tuning complexity relative to AdamW, raising questions about ease of adoption and robustness across settings.

**Additional Comments:**

The topic is relevant, but the contribution currently feels very incremental: IVON is existing work and the proposed selector is a simple heuristic. The experiments are suggestive but lack deeper analysis and deployable evaluation protocols.

**Audience:**

Yes

**Audience Explanation:**

The topic is of broad interest to the TMLR audience, and the results suggest that variational training can transfer effectively to multimodal settings and potentially improve reliability. That said, the current algorithmic design and analysis remain somewhat intuitive, and the paper could do more to provide deeper (theoretical/mathematical) insight into why the approach improves uncertainty estimates and under what conditions it may fail.

**Broader Impact Concerns:**

Abstention improves safety but may introduce disparate coverage across groups and shift risk/workload to downstream humans. If trained on user-provided images/questions, the system also raises privacy concerns around sensitive visual data handling and potential memorization, and multi-sample inference increases compute/energy costs.

**Claims And Evidence:**

Yes

**Claims Explanation:**

Overall, most claims are supported by experimental evidence on standard datasets using established metrics for calibration and selective prediction.

**Requested Changes:**

Clarify novelty and positioning: Be explicit that the primary novelty is applying IVON to VLMs plus the selective-prediction selector, and more clearly distinguish what is inherited from IVON versus what is new in this work.

Quantify training overhead: Report wall-clock training time and memory comparisons for AdamW vs. IVON.

Emphasize results that choose the selection threshold $\gamma$ on a validation set, and/or report how well a validation-chosen  $\gamma$ transfers to test risk/coverage, since test-time threshold selection can overstate deployability.

Finally, presentation and claims should remain appropriately conservative regarding real-world reliability improvements.

---

> ### Author Response · Authors · 2026-01-23
>
> We thank the reviewer for their insights and valuable feedback. We appreciate the reviewer’s recognition of our work as targeting a “clear and timely problem” and of our choice of baselines as “useful and practically relevant.” Below, we address the concerns. In the newly uploaded PDF (both main paper and supplementary material), all changes have been marked in blue.
>
> **"The novelty feels primarily empirical/system-level rather than methodological."**
>
> - Firstly, we would like to refer to the criteria set by the TMLR guidelines, which state that (a) "novelty of the studied method is not a necessary criterion of acceptance" and (b) that "The claims made in the submission [have to be] supported by accurate, convincing and clear evidence". Regarding the newly proposed selector, we strongly believe that Tables 4 and 5 provide this evidence for our claim that it consistently outperforms the previous standard of sample averaging in high-stakes selective prediction.
> - Additionally, we have added a paragraph in Section 4.3 that explains our method from a Bayesian credible-interval perspective (in short: it naturally penalizes predictions with high epistemic uncertainty, and makes a conservative confidence prediction such that there is roughly an 84% posterior probability that the true class probability exceeds this value). Picking 1$\sigma$ is an empirical choice that balances conservatism against practical utility. Regarding the application of IVON: while the optimizer itself is established, applying it to large-scale multimodal architectures required non-trivial adaptation. Our work represents the first demonstration that variational learning can be effectively applied to large VLMs and vision-language tasks without sacrificing accuracy, which we validated across multiple architectures and datasets.
>
> **"IVON introduces additional hyperparameters and tuning complexity relative to AdamW."**
>
> - We agree that practitioners accustomed to AdamW may need some time to adapt to the tuning required for IVON. One purpose of this work is to show that, despite this hurdle at first, one can identify correlations and sensible ranges for hyperparameters that work well for VLMs. Please see Appendix A for a discussion.
>
> **"Clarify novelty and positioning more explicitly."**
>
> - In response to the reviewer's request, we have clarified our phrasing in the introduction and the discussion. We look forward to discussing this further with the reviewer.
>
> **"Quantify training overhead with wall-clock time and memory comparisons."**
>
> - We have extended Section B of the Appendix, which reports wall-clock training time, by adding new figures that compare peak GPU memory usage for AdamW vs. IVON across different models and training batch sizes, as well as data on training and inference time for different numbers of MC samples.
>
> **"Emphasize validation-set threshold selection and report generalization to test."**
>
> - We emphasize that in all our results tables, we show results for Effective Reliability ($\Phi_c$) for multiple cost values. As discussed in Section 5.2, we chose to present this metric specifically because the standard C@R metric has the weakness of requiring test-set threshold selection, and Effective Reliability remedies this. We opted to still show C@R alongside Effective Reliability because it is standard in the field of Selective Prediction. We note that the general trends of C@R and $\Phi_c$ are very similar, particularly when errors are similarly costly, as for C@1 and $\Phi_{100}$.
> - Additionally, we have added results on threshold generalization (Appendix, new Section H) and found that, for both VarVQA and the baselines, the test risk closely matches the desired validation risk across all models and datasets. Test coverage, in turn, is closely correlated to test risk. A high test risk implies high test coverage, and vice versa, which complicates the evaluation and comparison of methods. We discussed this in Section 5.2.
>
> **"Presentation and claims should remain appropriately conservative regarding real-world reliability."**
>
> - Could the reviewer point us to specific parts of the paper that could benefit from this?
>
> **"Abstention raises concerns about disparate coverage, privacy, and computational costs."**
>
> - Privacy concerns around sensitive visual data and memorization apply broadly to all vision-language models, regardless of the training method used. Our variational approach does not introduce unique privacy risks compared to standard AdamW training—both methods learn from the same data and face identical privacy considerations.
> - Regarding computational costs: Trading computation for improved reliability is well-established in uncertainty estimation (e.g., Ensembling, MC Dropout, Self-Consistency, Verifiers). For high-stakes settings where errors are costly, multi-sample inference provides substantial gains that justify the computational investment.

---

> > ### Author Response · Authors · 2026-01-23
> >
> > **"The experiments lack deeper analysis and deployable evaluation protocols."**
> >
> > - Regarding evaluation protocols: We have documented our experimental setup in Section 5.1 and provide complete hyperparameter details in Appendix A, including correlations and working regimes to aid future practitioners. We will also release our code upon acceptance to ensure full reproducibility. Regarding the analysis: We have focused our evaluation on high-stakes selective prediction scenarios (low error tolerance, high error costs), which prior work has largely neglected despite these being the regimes where (a) differences between methods are most pronounced and (b) overconfident errors are most harmful to real-world deployment. We provide comprehensive evaluation across multiple dimensions: calibration, selective prediction at various risk levels, sample efficiency, and robustness to distribution shift. We would appreciate it if the reviewer could point to specific aspects of the experiments or analysis that they feel are lacking, so we can address these concerns more directly.

---

### Review · Reviewer_GBnG · 2025-12-11

**Summary Of Contributions:**

Positive Aspects

	-- New Application of Bayesian Methods: This paper successfully applies a variational Bayesian approach (the IVON optimizer of Shen et al., 2024) to large Vision-Language Models (VLMs) for Visual Question Answering. The Variational VQA (VarVQA) framework learns a distribution (mean and variance) over model weights, enabling intrinsic uncertainty estimation without additional model components. The authors show that variational training can be applied to a state-of-the-art VQA transformer (BEiT-3) with no loss in accuracy or significant training overhead. Further, the central strength of the work lies in significantly improved selective prediction under strict error tolerances.VarVQA also consistently outperforms Monte Carlo dropout when both use the same number of inference samples. VarVQA produces far more trustworthy VQA systems, especially in scenarios where mistakes are very costly. Additionally, the paper shows that their BPA selector outperforms conventional max-prob on the variational model, especially in high-stakes regimes.

	-- Improved Calibration and Reliability: Models trained with VarVQA are markedly better calibrated than those with standard training. The paper shows VarVQA achieves a lower Expected Calibration Error (ECE) than even post-hoc calibration techniques like temperature or vector scaling. For example, VarVQA yields a smaller ECE on VQAv2 compared to a vector-scaled baseline, while matching the calibration performance of Monte Carlo dropout. This is impressive, as calibration typically requires extra tuning, whereas VarVQA provides well-calibrated probabilities out-of-the-box. Empirically, just using the mean weights of a VarVQA model (no sampling) already produces lower ECE and higher selective accuracy than a conventional model with the same inference budget.

	-- Risk-Averse Selection Function: Another positive aspect is the introduction of a simple yet effective risk-averse selection strategy (g_BPA  that leverages the output variance. Instead of using only the mean predicted probability for confidence (as in standard max-prob or predictive averaging), the authors subtract the model’s uncertainty (standard deviation) for the top answer which penalizes answers with high posterior variance. The paper shows that this BPA selector outperforms conventional max-prob on the variational model, especially in high-stakes regimes.

	-- The study is mostly empirically rigorous except for limited baselines. The authors evaluate across multiple datasets – VQAv2 for in-domain VQA, NLVR2 for visual reasoning, and a mixture of VQAv2 with AdVQA for a distribution shift scenario. They test multiple different model scales (from a smaller ViLT model to BEiT-3 base and large), demonstrating that VarVQA’s benefits hold across architectures and sizes, on key standard metrics.

Negative Aspects

	-- Baseline Comparisons could be expanded: While the paper compares against Monte Carlo dropout and standard training, it omits some other competitive uncertainty estimation approaches. In particular, it would strengthen the work to compare against deep ensembles, which are a known standard for uncertainty (albeit they are costly but for uncertainty calibration this might be somewhat less relevant — nonetheless the tradeoff would be interesting to the community). The authors also confess that ensembling can yield   excellent calibration but is resource-intensive; however, a small-scale ensemble experiment (even with fewer models or smaller backbones) could quantify how VarVQA measures up to that sort of like an “upper bound” — I think this tradeoff would add a lot of value to the paper. Similarly, other prior VQA-specific selective prediction methods (e.g. the head-based confidence model of Whitehead et al., 2022, or the LLM+vision-tool approach of Srinivasan et al., 2024) are discussed in related work but not directly compared in experiments. This raises the question if the paper’s improvements are mainly due to the threshold selection mechanism  ... rather than just the Bayesian training itself – a direct empirical contrast with those methods (or a combination thereof) would bee illuminating.

**Audience:**

Yes

**Audience Explanation:**

the paper makes a noteable contribution of using bayesian methods for multimodal calibration which i feel several tlmr audiences will be interested in.

**Claims And Evidence:**

Yes

**Claims Explanation:**

i would say mostly yes, but the set of baselines could have been expanded (see strengths/weaknesses above)

**Requested Changes:**

-- More baselines (see weaknesses above)

Questions:
-- The authors cite past work on the shortcomings of variational Bayes for large networks (e.g. over-pruning effects), yet the paper does not deeply discuss whether such issues arose in their experiments. Could the authors clarify ? Did the authors notice any cases of the variational model becoming overconfident or underconfident due to the limitations of the approximation? A discussion of potential failure modes (e.g. if the variances collapse prematurely e.g. see Coker et al. 2022 on wide BNNs) would help understanding of the method’s robustness.

-- Behavior on Unanswerable or Out-of-Distribution Questions: How does VarVQA handle cases where the question cannot be answered from the image? Intuitively, a well-calibrated model should abstain ? Did the authors notice something like this ?

-- The method uses a diagonal Gaussian posterior for practicality. Do you think a more expressive posterior (like a full-rank covariance or a mixture) would significantly improve results, or is mean-field sufficient here? In other words, are the current bottlenecks more due to, say, data/label noise and model capacity, or due to the approximate inference? If you have insights or evidence (perhaps from the Foong et al. 2020 expressiveness perspective), it would be great to discuss whether pursuing richer posteriors might yield marginal gains or major improvements in selective performance.

-- Why did you particularly choose this selective technique instead of something like confidence head?

Minor: there are a couple of typos:

Calibraion– Missing  “t”

“BEiT-3 s near-SOTA”

“framework Chow (1957)” —> “framework by Chow (1957)” or “framework (Chow, 1957)” or may be the authors wanted to use \citep instead of \citet

Reference: “Weight uncertainty in neural network.” —> “neural networks”

---

> ### Author Response · Authors · 2026-01-23
>
> We thank the reviewer for their insights and valuable feedback. We are pleased that the reviewer finds it "impressive that [...] VarVQA provides well-calibrated uncertainties out-of-the box” and that our work is "empirically rigorous". Below, we address the concerns. In the newly uploaded PDF (both main paper and supplementary material), all changes have been marked in blue.
>
> **"Could baseline comparisons be expanded to include deep ensembles?"**
>
> - We agree that deep ensembles represent a strong baseline for uncertainty estimation. Following the reviewer's suggestion, we have added Table 3 to the main paper, comparing three-model deep ensembles on VQAv2. We note that ensembling can be applied on top of any training method, including VarVQA. Notably, VarVQA (single model) outperforms the AdamW deep ensemble on high-stakes metrics, and VarVQA ensembles further improve performance over single-model VarVQA. We provide a comprehensive ensemble study in the new Appendix Section F, with full results on VQAv2 (Table 11) and NLVR2 (Table 12). On NLVR2, results are more mixed, which we attribute to the smaller test set size (5,000 examples vs. 100,000 for VQAv2), leading to higher variance across seeds, particularly for high-stakes metrics that are sensitive to individual overconfident predictions. We are working to expand the ensemble size to strengthen these conclusions.
>
> **"Why not compare directly with other VQA-specific selective prediction methods?"**
>
> - We appreciate this suggestion. Our work focuses on demonstrating that variational training can improve the intrinsic reliability of vision-language models, rather than claiming superiority over all possible selective prediction approaches. These external selector methods represent fundamentally different design philosophies: they train an additional component on top of a frozen (and potentially unreliable) predictive model, requiring task-specific training phases and additional parameters, while the underlying model remains miscalibrated and does not inherently "know" its own uncertainty. In contrast, VarVQA improves the model's intrinsic confidence estimates—uncertainty estimation, calibration, and selective prediction work well out of the box without requiring separate training or additional components.
> - Still, following the reviewer's request, we have added a comparison with the Whitehead et al. (2022) selector approach in Appendix Section H (Table 15). Despite the additional computational cost and explicit training phase for the selector, it provides only small improvements over VarVQA.
> - Regarding the Srinivasan et al. (2024) vision-tool approach, a direct comparison is not meaningful due to fundamental differences in the evaluation protocols. Their method specifies a desired risk level in advance, which can and does lead to test risks that deviate substantially from the target. This makes fair comparison difficult: if their method yields a test risk higher than intended, it naturally inflates coverage, creating an unfair advantage. Our evaluation protocol instead follows standard practice.
>
> **"Did you observe pathologies like over-pruning or variance collapse?"**
>
> - Thank you for this question. IVON does not appear to suffer from the pathologies observed in earlier variational methods. Specifically, regarding over-pruning (Trippe and Turner, 2018), we investigated the learned parameter magnitudes and found minimal differences between AdamW and IVON. For example, on VQAv2-finetuned BEiT-3, IVON has 4% of weights with magnitude below 1e-3 compared to 3% for AdamW—an insignificant difference that suggests no over-pruning occurs. Regarding variance collapse and data-ignoring (Coker et al., 2022): We do not observe the failure modes associated with wide mean-field BNNs. Parameter variances steadily increase during training rather than collapsing, and the posterior meaningfully deviates from the prior, indicating the model is learning from the data. Our competitive accuracy (matching AdamW) and improved uncertainty estimates further confirm that IVON learns meaningful posterior distributions in our multimodal setting.
>
> **"How does VarVQA handle unanswerable or out-of-distribution questions?"**
>
> - Our models have been fine-tuned on the specific datasets VQAv2 and NLVR2, and thus cannot be expected to be perfect abstainers outside their distributions. While we did not specifically investigate unanswerable questions, we observed that VarVQA's reliability in OOD settings remains ahead of baselines like AdamW, but it eventually breaks down when the test distribution is too far from the training distribution. This would likely be the case for unanswerable questions, so while VarVQA could be expected to be somewhat less overconfident than its AdamW alternative, we do not expect an abstention miracle there.

---

> > ### Author Response · Authors · 2026-01-23
> >
> > **"Would a more expressive posterior significantly improve results?"**
> >
> > - We do believe that a more expressive posterior should improve the performance. For instance, using an isotropic Gaussian in IVON necessarily yields worse results (see NeurIPS 25, Ghosh et al., "Variational Learning Finds Flatter Solutions at the Edge of Stability"), so we believe moving from a diagonal covariance to a full covariance should improve results. However, for now, there are no practical alternatives to IVON for such posteriors, so it is difficult to say whether this observation is correct. We have added a sentence about this in the discussion section.
> >
> > **"Why this selective technique instead of a confidence head?"**
> >
> > - Our goal was not only to improve Selective Prediction on VQA, but also to specifically investigate the potential of Bayesian methods to improve the reliability of large multimodal models. In addition (as mentioned above), a confidence head requires a task-specific training phase, whereas VarVQA can deliver calibrated uncertainties "out-of-the-box".
> >
> > **"Typos"**
> >
> > - We thank the reviewer for spotting several typos, which we have corrected. Regarding the Blundell (2015) paper, while we also think the grammatically correct version should be "networks", the original paper title is "Weight uncertainty in neural network".

---

### Review · Reviewer_X9RQ · 2026-01-16

**Summary Of Contributions:**

This paper proposes Variational VQA (VarVQA), which introduces variational Bayesian learning into vision-language models by leveraging the IVON optimizer to learn a posterior distribution over model parameters. This enables intrinsic uncertainty modeling without sacrificing accuracy or requiring additional model components. Through extensive experiments on visual question answering and visual reasoning tasks, the authors show that VarVQA substantially improves calibration and selective prediction performance, with particularly strong gains in high-stakes settings where error tolerance is low. Moreover, the proposed approach is more sample-efficient than MC Dropout and demonstrates increased robustness under distribution shift. Overall, this is a novel and promising work that leverages variational inference to advance multimodal language models.

**Audience:**

Yes

**Audience Explanation:**

Yes. The proposed method improves the reliability of multimodal large language models for VQA, a very interesting and relevant topic in ML domain.

**Broader Impact Concerns:**

The paper aims to improve the reliability and trustworthiness of multimodal large language models through better uncertainty estimation and selective prediction. The proposed approach is intended to reduce overconfident errors rather than enable new capabilities, and no significant negative societal or ethical impacts are foreseen.

**Claims And Evidence:**

Yes

**Claims Explanation:**

Yes. The paper provides comprehensive experiments across multiple VLM architectures, datasets, and evaluation metrics, showing consistent improvements in calibration, selective prediction and sample efficiency over baselines such as AdamW and MC Dropout.

**Requested Changes:**

- Although the BPA selector $\mu - \sigma\$ is empirically effective, can the author provide a more intuitive explanation of why it works?
- ViLT and BEiT-3 are not the most commonly used VLMs today; why the author choose these model, and do you considering more modern VLM models such as LLaVA-1.6, or Qwen2.5-VL, etc?
- Section 5.3 seems did not include the discussion of Figures 3(b), 3(c), and 3(d).

---

> ### Author Response · Authors · 2026-01-23
>
> We thank the reviewer for their insights and valuable feedback. We thank the reviewer for acknowledging that our work is "novel and promising" and that our "comprehensive experiments" were praised. Below, we address the concerns. In the newly uploaded PDF (both main paper and supplementary material), all changes have been marked in blue.
>
> **"Although the BPA selector is empirically effective, can the author provide a more intuitive explanation of why it works?"**
>
> - We have added a paragraph in Section 4.3 that features an explanation from the viewpoint of (Bayesian) credible intervals. In short, $\mu - \sigma$ can be interpreted as approximately the 16th percentile of the predicted class probability distribution, meaning there is roughly an 84% posterior probability that the true class probability exceeds this value. This provides a principled, conservative estimate that naturally penalizes predictions with high epistemic uncertainty—i.e., cases where the model should abstain. Such a risk-averse approach is well-suited to high-stakes selective prediction where overconfident errors are costly. The 1$\sigma$ choice empirically balances conservatism against practical utility: more aggressive bounds (e.g., $\mu- 2\sigma$) can lead to excessive abstentions, while using only $\mu$ ignores the uncertainty information.
>
> **"ViLT and BEiT-3 are not the most commonly used VLMs today; why the author choose these model, and do you considering more modern VLM models such as LLaVA-1.6, or Qwen2.5-VL, etc?"**
>
> - While our method is architecture-agnostic and can be applied to any VLM, we chose ViLT and BEiT-3 because they are well-established VQA architectures with official implementations and VQA fine-tuning protocols, enabling controlled comparisons. Importantly, our uncertainty quantification method is architecture-agnostic and applies to any VLM. Newer models like LLaVA-1.6 and Qwen2.5-VL are also good candidates, but they are often instruction-tuned chat models rather than task-specific VQA models. This can make controlled comparison tricky, but we may consider such extensions in the future.
>
> **"Section 5.3 seems did not include the discussion of Figures 3(b), 3(c), and 3(d)."**
>
> - We thank the reviewer for spotting this. We have modified Section 5.3 accordingly.

---

### Decision · Action_Editor_GvSo · 2026-03-11

**Recommendation:** Accept as is

**Additional Comments:**

Regarding Reviewer ZqNN's concerns about incrementality: TMLR's evaluation criteria explicitly state that novelty of the studied method is not a necessary criterion for acceptance.

**Audience:**

Yes

**Audience Explanation:**

Two of three reviewers affirm audience interest (I concur). That fulfills the "at least some individuals" criterion.

**Claims And Evidence:**

Yes

**Claims Explanation:**

All three reviewers agree that the claims are supported by experimental evidence. The authors have responsed to the reviews and addressed the comments.

(Personally, it always breaks my heart when I see a paper compute the variance of samples from the probability simplex when the mutual information would be a much more natural metric, but that's just me, it seems.)